# *Phytoene synthase 1* (*Psy-1*) and *lipoxygenase 1* (*Lpx-1*) Genes Influence on Semolina Yellowness in Wheat Mediterranean Germplasm

**DOI:** 10.3390/ijms21134669

**Published:** 2020-06-30

**Authors:** Roberto Parada, Conxita Royo, Agata Gadaleta, Pasqualina Colasuonno, Ilaria Marcotuli, Iván Matus, Dalma Castillo, Adriano Costa de Camargo, Jorge Araya-Flores, Dolors Villegas, Andrés R. Schwember

**Affiliations:** 1Departamento de Ciencias Vegetales, Facultad de Agronomía e Ingeniería Forestal, Pontificia Universidad Católica de Chile, Santiago 306-22, Chile; Roberto.paradasalazar@gmail.com (R.P.); adrianoesalq@gmail.com (A.C.d.C.); j.araya.f@gmail.com (J.A.-F.); 2Sustainable Field Crops Programme, Institute for Food and Agricultural Research and Technology (IRTA), 25191 Lleida, Spain; Conxita.Royo@irta.cat (C.R.); dolors.villegas@irta.cat (D.V.); 3Department of Agricultural and Environmental Science, University of Bari Aldo Moro, 70121 Bari, Italy; agata.gadaleta@uniba.it (A.G.); pattybiotec@yahoo.it (P.C.); i.marcotuli@gmail.com (I.M.); 4Instituto de Investigaciones Agropecuarias (INIA), Centro Regional de Investigación Quilamapu, Chillán 426, Chile; imatus@inia.cl (I.M.); dalma.castillo@inia.cl (D.C.); 5Laboratory of Antioxidants, Nutrition and Food Technology Institute, University of Chile, Santiago 7810000, Chile

**Keywords:** durum wheat, grain carotenoids, semolina yellowness, landraces, *phytoene synthase 1*, *lipoxygenase 1*

## Abstract

*Phytoene synthase 1* (*Psy1*) and *lipoxygenase 1* (*Lpx*-1) are key genes involved in the synthesis and catalysis of carotenoid pigments in durum wheat, regulating the increase and decrease in these compounds, respectively, resulting in the distinct yellow color of semolina and pasta. Here, we reported new haplotype variants and/or allele combinations of these two genes significantly affecting yellow pigment content in grain and semolina through their effect on carotenoid pigments. To reach the purpose of this work, three complementary approaches were undertaken: the identification of QTLs associated to carotenoid content on a recombinant inbred line (RIL) population, the characterization of a Mediterranean panel of accessions for *Psy1* and *Lpx-1* genes, and monitoring the expression of *Psy1* and *Lpx-1* genes during grain filling on two genotypes with contrasting yellow pigments. Our data suggest that *Psy1* plays a major role during grain development, contributing to semolina yellowness, and *Lpx-1* appears to be more predominant at post-harvest stages and during pasta making.

## 1. Introduction

Wheat is one of the most important cereal crops worldwide [1,2], as about 725 million tons of grain are produced globally every year [3]. Durum wheat (*Triticum turgidum* L. ssp. *durum*) is a tetraploid species, with genomes A and B, used in a variety of food products, including couscous and bulgur, but mostly semolina, the raw material for pasta manufacturing [4,5,6]. Durum represents about 8% of the total wheat cultivated area and 5% of global wheat production [7]. In the twentieth century, major breeding programs were focused on improving the durum productivity traits of wheat, such as grain yield and biotic and abiotic stress resistance. In this century, the attention on food quality over quantity has switched the research aims to increasing wheat nutritional value, estimated through different parameters like protein content, water absorption, and flour color. The latter is due to carotenoid pigments, which have an enormous importance for nutritional value in human health [8].

Yellow pigment content (YPC) is one of the most important quality features of durum wheat grain [6,9], due to the positive correlation existing between yellowness and pasta quality [10], which is associated with a higher consumer acceptance [2]. For this reason, improving YPC is a main goal in most durum wheat breeding programs [2,11], particularly considering that the competition in the pasta market has made this trait even more important [12], especially after the legal ban of the use of artificial coloring in pasta production in certain countries in Europe [13], which has strengthened the role of durum wheat breeding programs for YPC enhancements.

The carotenoid content in the wheat grain is mainly composed of lutein and small amounts of zeaxanthin and β-cryptoxanthin [14,15]. Lutein is a compound that contributes to the organoleptic quality of pasta (yellowness) [14]. A high carotenoid content in pasta enhances its nutritional value due to the cell membrane protective role of carotenoids against oxidative damage [4], by reducing the effective concentration of free radicals in the cells [9,16]. Grain and semolina yellowness depend on several factors, including the endogenous carotenoid content of kernels, grinding rate, the pasta-making conditions, and the oxidative degradation of enzymes, in which lipoxygenases (LOX) play a major role [9,17]. Thus, breeding programs should aim to produce cultivars with high endogenous carotenoid pigments and low oxidative activity [10].

Although grain yellowness is genetically controlled [9,18,19], it is a very complex polygenic trait [20,21]. Quantitative trait loci (QTLs) for YPC have been found across all wheat chromosomes. A complete and comprehensive review of the QTLs and genes (with the specific accession numbers) that affect YPC is reported in Colasuonno et al. [7].

The enzyme phytoene synthase (PSY) catalyzes the rate-controlling step in the synthesis of phytoene from geranylgeranyl diphosphate. *Psy1-A1* and *Psy1-B1* encoding *phytoene synthase 1* are major QTLs for YPC in chromosomes 7A and 7B, respectively [22]. There is considerable information regarding allelic variation in these genes related to grain yellowness variation in wheat. Several authors had reported and developed markers in both *Psy1-A1* and *Psy1-B1* that are able to select cultivars for high yellowness levels in both durum and common wheat [22,23,24,25,26,27,28]. Three allelic variants at *Psy1-A1* have been reported, including *Psy1-A1a* (low semolina yellowness), *Psy1-A1l* (intermediate yellowness), and *Psy1-A1o* (high yellowness) [29]. However, a previous study of our group that involved 155 Mediterranean landraces and 20 modern cultivars of diverse origin associated the presence of the allele *Psy1-A1l* with the highest values of semolina yellowness and the presence of *Psy1-A1a* with the lowest values [30]. In that study, no differences between *Psy1-B1a* and *Psy1-B1b* were identified as being related to semolina yellowness.

Nevertheless, a high carotenoid content and a high endosperm yellowness do not guarantee high yellowness content in pasta products. During pasta processing, the oxidative degradation of carotenoids can occur, leading to the bleaching of the end-products. Even though there are several enzymes contributing to this effect, including peroxidases and polyphenol oxidases, lipoxygenases are the main players [6,7,9,11,31]. Lipoxygenase enzymes catalyze the addition of molecular oxygen across the cis, cis-1,4 pentadiene system to produce the corresponding hydroperoxides [32]. In plants, lipoxygenases are found in leaves, seedlings, and seeds. LOX activity produces reactive oxygen species (ROS), originating from fatty acid oxidation, and these radicals can produce the oxidation and degradation of carotenoids [10]. In durum wheat, there are different lipoxygenase genes and alleles contributing to the variation in pasta yellowness [11,32]. Hessler et al. [31] sequenced in durum wheat several fragments of *lipoxygenase-1*, which was reported to be responsible for LOX activity in barley seeds [33], and based on the similarity to the *LoxA* gene between the two species, these sequences were assigned to the *Lpx-1* locus in durum wheat [31]. De Simone et al. [34] reported different levels of *Lpx-1* and *Lpx-3* transcripts at maturity between cultivars with contrasting LOX activities, meanwhile, *Lpx-2* transcripts were absent at this stage. The *Lpx-B1* locus is located on the short arm of chromosome 4B [16,31,32] and five related genes and allele sequences have been reported, *Lpx-B1.1a* (Genbank HM126466), *Lpx-B1.1b* (Genbank HM126468), and *Lpx-B1.1c* (Genbank HM126470) for the *Lpx-B1.1* locus and *Lpx-B1.2* (Genbank HM126467) and *Lpx-B1.3* (Genbank HM126469) [4,31,32]. QTL analyses in durum wheat showed that 36–54% of the variation in LOX activity is attributable to *Lpx-B1* [4,31]. Additionally, Verlotta et al. [32] identified three different combinations between the alleles for *Lpx-B1.1* and the *Lpx-B1.2* and *Lpx-B1.3* genes, named haplotypes I (*Lpx-B1.1b* and *Lpx-B1.3*), II (*Lpx-B1.1a* and *Lpx-B1.2*), and III (*Lpx-B1.1c* and *Lpx-B1.2*), in which haplotypes I, II, and III showed high, intermediate, and low levels of functional *Lpx-B1* transcripts and enzymatic activity, respectively. Thus, understanding LOX activity in mature durum wheat kernels is critical for semolina yellowness and end-products derived from this cereal.

This study was conducted to identify new allele variants and/or allele combinations significantly affecting YPC in semolina through their effect on the synthesis and degradation of carotenoid pigments in grain and semolina. For this purpose, three complementary approaches were undertaken: the identification of QTLs associated with carotenoid content on a recombinant inbred line (RIL) population, the characterization of a Mediterranean panel of accessions for *Psy1* and *Lpx-1* genes, and monitoring the expression of *Psy1* and *Lpx-1* genes during grain filling in two genotypes with contrasting YPC.

## 2. Results

### 2.1. Detection of QTLs for Carotenoid Content in the RIL Population

#### 2.1.1. Phenotyping

The ANOVA of the yellow index (YI) data assessed in the RIL population revealed that all factors in the analysis (genotype (G), environment (E), and G × E interaction) were statistically significant (*p* < 0.01, data not shown). The parental lines differed significantly in YI values in the three years, with *cv.* “Saragolla” showing consistently higher values than “02-5B-318” (Table 1). The RIL values for YI appeared normally distributed and showed ranges of 5.38, 6.39, and 5.86 in 2015, 2016, and 2017, respectively (Table 1). The broad-sense heritability of YI ranged from 0.85 to 0.88 in the three testing years, highlighting that the phenotype was largely due to a genotypic effect (Table 1).

#### 2.1.2. QTL Analysis and Identification of Carotenoid Genes

As a starting point for the identification of QTL regions, the genetic map developed for the RIL population by Giancaspro et al. [35] was used as a reference for the analysis. The composite interval mapping (CIM) method was employed for QTL analysis. Putative QTLs for YI are listed in Table 2. Eight QTLs for YI (with LOD > 3) were mapped on chromosomes 2B (one), 4A (one), 4B (three), 5A (two), and 7A (one). The percentage of phenotypic variation (*R^2^*) for YI explained by individual QTLs ranged from 10% to 59%. The major QTL was detected on chromosome 2B, flanked by the markers IWB12724 and IWB11333 (IWB32245 as the closest marker). This QTL was significant within the three years and across them and explained from 30% to 59% of YI variations. The QTLs on chromosomes 5A (interval IWA1258-IWB72888) and 7A (interval IWB73689-IWB25891) were statistically significant in all cases, accounting from 12% to 21% of the phenotypic variation. The QTL on chromosome 4B (IWB6397-IWB44213) was significant in two environments, accounting for 13% of YI variation. The QTLs on chromosomes 4A (IWB12722-IWB14501), 4B (IWB73832-IWB11928; IWB71402-IWB53932), 5A (IWB65257-IWB35711), and 7A (IWB49295-IWB8841) were significant in one environment with *R^2^* values ranging between 10 and 20%. Table 2 summarizes the location of QTLs on the genetic map, their LOD scores, and the closest markers flanking the region with the respective associated marker.

In order to verify the possible presence of candidate genes for the yellow index trait, all single nucleotide polymorphism (SNP) markers present in the detected QTL regions were studied. These SNP markers, mapped by Giancaspro et al. [35], were subjected to BLASTn analysis (based on percentage identity) to verify that the sequences have a high percentage of homology with the biosynthesis genes of the carotenoid pigments. All gene sequences used as queries were derived from Colasuonno et al. [7]. The analysis allowed for the identification of the lipoxygenase gene (*Lpx*) in the QTL region on chromosome 4B (IWA103 marker located in the 4B-2 QTL region, Table 2), confirming the key role of this chromosome in trait control. For the other key genes involved in the catabolic and biosynthetic pathway of the carotenoid pigment, no polymorphic markers were present in the analyzed regions.

#### 2.1.3. Expression Profile of Lpx Genes on Leaves

The relationships between *Lpx* and yellow index were analyzed in the leaf tissue of the bread wheat accession “02-5B-318” and the durum wheat *cv.* “Saragolla”, characterized by low and high YIs, respectively. The *Lpx* genes mapped on chromosomes 4A, 4B, 5A, and 5B were considered in the expression study in order to analyze each homoeologous *Lpx* gene. Total RNA was extracted from kernels, and quantitative real-time PCR (qPCR) was carried out with genome-specific primers. Significant statistical differences were observed in the three homoeologous forms in the two parental lines. High expression levels and significantly different expression values (*p* < 0.05, *t*-test) were detected between genotypes “02-5B-318” and “Saragolla” (0.67 and 1.58, respectively) for the *Lpx* gene on chromosome 5B, indicating that the *Lpx* allele present in *cv*. “Saragolla”, the parent with a a high YI, was more active in the accumulation of yellow pigment that the allele present in “02-5B-318”, the parent with a low YI (Figure 1). Furthermore, significant differences (*p* < 0.05) were also observed for *Lpx* genes located on homologous group 4, showing the positive contribution to the trait of the “02-5B-318” parent, with normalized expression data of 1.0 for both the A and B isoforms compared to 0.32 and 0.47 for the 4A and 4B genes of *cv.* “Saragolla”.

### 2.2. Characterization of the Mediterranean Panel for Psy1-A1 and Lpx-B1

A previous study of our team identified the allele composition of the *Psy1-A1* gene as a reliable indicator of semolina b* value [30]. To gain more insight into the genomic background of this gene, allele variants of *Psy1-A1* were assessed in the whole population, analyzed herein, and the results are shown in Table 3. Similar allele frequencies were found for the alleles *Psy1-A1a* and *Psy1-A1o*, but *Psy1-A1l* predominated (Table 4). Semolina b* values were significantly lower for genotypes harboring the *Psy1-A1a* allele in comparison to the ones carrying the alleles *Psy1-A1l* or *Psy1-A1o*, which showed similar values (Table 4).

Primer pairs outlined in Table 5 allowed the identification of five combinations (haplotypes) between allele variants of *Lpx-B1.1, Lpx-B1.2*, and *Lpx-B1.3* genes (Table 3). Three of them correspond to those previously described by Verlotta et al. [32], i.e., haplotypes I (*Lpx-B1.1b* and *Lpx-B1.3*), II (*Lpx-B1.1a* and *Lpx-B1.2*), and III (*Lpx-B1.1c* and *Lpx-B1.2*), but two new ones with a low frequency in the population, named haplotypes IV and V following the nomenclature of Verlotta et al. [32], were found for the first time in the present study (Table 6). Haplotype IV, which showed the combination of *Lpx-B1.1b* and *Lpx-B1.2*, was identified in three landraces from Spain, Cyprus, and Morocco, whereas haplotype V, formed by the combination of *Lpx-B1.1a* and *Lpx-B1.3*, was detected in four landraces, two French, one Syrian, and one Spanish (Table 3). Alignments were made between the obtained amplicons and the corresponding sequences for *Lpx-B1.1a*, *Lpx-B1.1b*, *Lpx-B1.1c*, *Lpx-B1.2*, and *Lpx-B1.3*, showing no presence of new genes or allele sequences within the population evaluated (data not shown).

Large variability was identified in haplotypes I and II, with b* values ranging from 12.38 to 25.78, and 10.32 to 23.49, respectively. These haplotypes were the most frequent within the Mediterranean panel (Table 6). The results of the ANOVA and the comparison of the mean b* values of the five haplotypes showed no statistical differences among them (Table 6). Interestingly, all modern genotypes harbored haplotype II (Table 3). When the genotypes carrying haplotype II were segregated in “haplotype II landraces” and “haplotype II modern cultivars”, significant differences appeared between them, as the latter encompassed the cultivars with higher levels of yellowness (Figure 2).

To further dissect the population into smaller groups and get more information on the combinations between allele variants at different loci, we named the combinations between the three *Psy1-A1* allele variants and the five *Lpx-B1* haplotypes identified in this study. Eleven different combinations were obtained (named 1 to 10), and since eight of the nine modern genotypes harbored combination 5, the modern/landrace distinction was applied to it (Table 7). As expected, groups carrying *Psy1-A1a* (1 and 4) showed the lowest b* values. Allele combinations 2, 4, and 5 harbored approximately 80% of the population. By separating genotypes carrying combination 5 into modern cultivars and landraces, modern cultivars exhibited the highest b* values, which were significantly higher than the b* values of landraces from allele combination 5. The highest levels of yellowness (b* values over 19.0) were recorded for allele combinations 3 and 8 and in modern cultivars, without significant differences between them (Table 7).

### 2.3. Expression Levels of Psy1 and Lpx-B1 Genes during Grain Filling in Genotypes with Contrasting YI

The transcription levels of *Psy1* and *Lpx-B1* genes during grain filling were assessed in two landraces that showed consistently high (DW028) and low (DW011) YI values in each of the three environments (Table 3). The results showed that DW011 had no statistical difference in *Lpx-B1* gene expression levels during the grain-filling period, and *Psy1* had a peak at 28 days post anthesis (DPA) with a subsequent decrease in its expression (Figure 3A). The high yellowness line (DW028) exhibited no statistical expression differences for the *Lpx-B1* gene, similarly to DW011, but *Psy1* showed an increase in expression from 14 to 49 DPA, peaking at 42 and 49 DPA (Figure 3B). When the *Psy1* expression levels between the low and high yellowness genotypes were compared, they did not show statistical differences until 42 and 49 DPA, when DW028 had 7.5 and 5.8 times higher expression than DW011, respectively (Figure 3C). In the case of *Lpx-B1*, DW011 had 1.8 and 2.1 times higher expression than DW028 at 14 and 21 DPA, respectively, they showed no differences at 28 and 35 DPA, and the trend reverted at 42 and 49 DPA, when DW028 had 9.3 and 9.9 times higher expression than DW011 (Figure 3D).

## 3. Discussion

Yellowness is a major trait determining durum wheat quality for pasta making purposes. *Psy1* and *Lpx-1* have been pointed out as key enzymes predominately involved in carotenoid synthesis and degradation, respectively. Several studies reported many QTLs for yellow index spread all over the wheat genome [7], but to our knowledge no previous research has been reported simultaneously analyzing the two genes, the diversity of genetic materials, and expression profiles. Among them, two different QTLs were mapped on the long arm of chromosome group 7, co-localized with the *phytoene synthase 1* (*Psy1*) and *aldehyde oxidase 3* (*AO3*) genes, respectively [23,37,38,39]. While the *Psy1* involvement in YPC has been deeply studied, the role of the *AO3* gene in carotenoid accumulation needs to be elucidated. AO isoforms are key enzymes for abscisic acid (ABA) biosynthesis [40,41]. The plant AO family is composed of proteins with a high similarity in sequences, but different subunit compositions and substrate preferences. AO isoforms have been largely characterized in *Arabidopsis* and are composed of four isoforms [42].

In the current study, the *Lpx* gene was studied in a RIL population developed by crossing two genotypes, “02-5BIL-318” and “Saragolla”, which consistently differ in their yellow coloring, thanks to its association with a QTL located on chromosome 4B. The localization on chromosome 4B of the *Lpx* gene was confirmed even in other research studies [32,43]. As reported from other authors [32,44,45], the linkage analysis highlighted not only the connection between this gene and the QTL, but also confirmed the key function of this gene in the oxidation of carotenoids. Thus, only for the isoforms mapped on chromosome group 4 was it deduced that the genotypes with a low content of carotenoid pigments were characterized by a greater catabolic activity. Carrera et al. [4] showed how the role of lipoxygenase on carotenoid degradation occurs in the process of pasta making and not during wheat grain development, which is in agreement with our own results.

The distribution of the *Lpx-B1* gene family and the *Psy1-A1* alleles were studied in a large durum wheat population comprising 128 Mediterranean landraces and modern cultivars, in order to link their presence and distribution to semolina yellowness.

The allelic variant *Lpx-B1.1c* was present in only two of the 128 genotypes analyzed in the current study, and they had very different levels of yellowness (DW059 and DW008, Table 3), even though both carried the *Psy1-A1* alleles that are related to high levels of carotenoid synthesis (Table 4, [29]), and they are supposed to have low levels of LOX activity [4,32].

Verlotta et al. [32] made an association between LOX activity and haplotype types on wholemeal extracts, simulating the pH conditions (pH 6.6) during pasta processing, in which lipoxygenases are expected to be most active. In addition, Carrera et al. [4] reported that between 8 and 16% of the total carotenoid content is lost due to LOX activity during pasta processing.

Previous studies have not found relationships between LOX activity and yellow color in semolina [31,34]. In these studies, it was reported that in the steps from whole meal to semolina, and from semolina to pasta, there is a marked reduction in YPC, especially in genotypes having high LOX activity. LOX activity apparently becomes more important at the time of wheat processing, not before, so it makes more sense not to have found a relationship between the presence of the alleles and b* values. Additionally, when LOX activity was studied with different pH, it was reported that in varieties with high LOX activity, there is apparently more than one active LOX isoform, since the decrease in activity as the pH increases is not as pronounced as for cultivars with low LOX activity. In addition, a well-defined peak of maximum LOX activity was observed, which decayed rapidly when altering the pH, suggesting that in these cases a single isoform is produced, determining the LOX activity in wholemeal durum wheat. The authors found a relationship between LOX activity in semolina and yellowness in pasta. The presence of transcripts is not entirely consistent with the LOX activity reported in their work, so post-transcriptional mechanisms could be playing a role in the modulation of LOX activity. Transcript levels also associated with LOX activity in certain durum wheat varieties may be due to *Lpx-1* and *Lpx-3*, while in others, only to *Lpx-1*. In addition, Verlotta et al. [32] reported the possibility of the existence of more allelic variants for *Lpx-B1.2* and *Lpx-B1.3* that were not detected in the current work with the specific primers used. Further, we can speculate that the differential *Lpx-1* gene expression identified during the vegetative stage in this study, but not during grain filling, probably may not affect YPC levels, but post-transcriptional mechanisms, durum wheat processing conditions, and/or other allelic variants for *Lpx-1* do influence semolina yellowness.

The expression levels of *Psy1* and *Lpx-B1* genes were analyzed during grain filling using two Mediterranean durum wheat genotypes with contrasting levels of yellowness compiled and purified at IRTA: DW011 (low yellowness; b*= 14.18 ± 1.29) and DW028 (high yellowness; b*= 21.71 ± 0.55). *Psy1* exhibited a peak of expression at 42 and 49 DPA in the high yellowness landrace that could result in higher levels of grain phenotypic yellowness (Figure 3A,B). In the present study, no expression differences during grain filling were encountered for *Lpx-B1* in both contrasting genotypes. The high yellowness genotype had 7.5 and 5.8 times higher expression levels for *Psy1* than the low yellowness landrace at 42 and 49 DPA (Figure 3C). Interestingly, our group previously studied 12 modern durum wheat genotypes that showed greater expression of *Psy1-A1*, which specifically peaked at 35 DPA. At this time, *Psy1-A1* was 21-fold more highly expressed in the high yellowness genotypes relative to the low yellowness genotypes [46]. The results of the present study suggest that not only is the particular allele of *Psy1* responsible for the determination of grain yellowness (*Psy1-A1l,*
Table 4), but its expression levels may also play a role in this trait (Figure 3). In addition, the higher expression levels of *Psy1* of the landrace studied occurred late in grain development, while the modern durum genotypes had their peaks earlier (35 DPA), which may distinctly influence grain yellowness.

Regarding DW011, it is associated with haplotype II for *Lpx-B1* (Table 3) and group 4 for *Lpx-B1/Psy1-A1* (mean b* value 16.22 ± 0.39; Table 7), while DW028 is categorized as haplotype I for *Lpx-B1* (Table 3) and group 3 for *Lpx-B1/Psy1-A1* (mean b* value 19.25 ± 0.61; Table 7). There may be a relationship between the expression levels for *Psy1* in the late grain filling stages and b* value. In the case of *Lpx-B1,* the DW011 had 1.8 and 2.19 times higher expression levels at 14 and 21 DPA than DW028, but this last genotype exhibited 9.36 and 9.97 higher expression levels at 42 and 49 DPA than the low yellowness landrace. Since *Lpx-B1* encodes for lipoxygenase, lower gene expression levels were expected in the high yellowness genotypes, in fact, if the ratio *Psy1/Lpx-B1* expression levels are compared at 49 DPA, the low yellowness genotype showed a ratio of 41.75 and the high yellowness genotype had a ratio of 24.17. Considering that the b* values for DW011 and DW028 are 14.18 ± 1.29 and 21.71 ± 0.55, respectively (Table 3), our result suggests that the expression of *Psy1* had a greater influence than *Lpx-B1* on b* values, but protein quantification and activity assays are required to draw more definitive conclusions.

Finally, high levels of carotenoid pigments in wheat kernels have important positive implications for human health since they are antioxidant compounds and precursors of vitamin A. Identifying the role of the main carotenoid genes in durum wheat and the specific alleles present in each cultivar can allow the development of superior cultivars through marker-assisted breeding programs. Molecular markers associated with *Lpx-1* are currently being effectively used for marker-assisted selection in durum wheat breeding programs to improve YPC in different parts of the world, including the United States [11] and Canada [2].

## 4. Materials and methods

### 4.1. Plant Material and Experimental Setup

A set of 135 F_6-7_ RILs was developed by the Department of Environmental and Territorial Sciences, University of Bari, Italy (DISAAT) through a single seed descendant (SSD) method, as described by Giancaspro et al. [35], and used for QTL analysis. The parents were the bread wheat accession “02-5B-318”, derived from the Chinese *cv.* “Sumai-3”, characterized by a low YI, and the durum wheat *cv*. “Saragolla”, characterized by a high YI. The parents and the RIL population were grown in Valenzano (Metropolitan City of Bari) for three years (2015, 2016, and 2017) using a randomized complete block design with four replications. Each plot consisted of 1-m rows, 30 cm apart, with four g of seeds sown in each plot and supplemented with nitrogen (10 g/m^2^).

A Mediterranean durum wheat panel, including 119 landraces and nine modern cultivars (provided by IRTA, Table 8), was used to assess the relationship between *Psy1-A1* and *Lpx-B1* genetic composition and YI. This panel, which is described in detail in Nazco et al. [47], was grown in Chile during the 2016–2017 and 2017–2018 cropping seasons in Chillán (37°09′02.53″ S; 72°01′04.35″ W) and during the 2016–2017 cropping season in Pirque (33°40′00″ S; 70°35′23″ E). Experimental designs were randomized complete blocks with three replicates and plots of three 1-m rows and 0.2-m inter-row spacing, planted at a seed rate of 220 kg/ha. Fertilizers were applied at a dose of 23 kg N/ha as urea (46% N), 27 kg N/ha + 69 kg P_2_O_5_/ha as diammonium phosphate (18% N, 46% P_2_O_5_), and 60 kg KCl/ha of potassium chloride (60% KCl) before sowing, and additionally 284 kg N/ha were applied as urea (46% N) at tillering. Plots were irrigated when necessary to prevent water deficit. Weeds, aphids, and fungal diseases were chemically controlled. Plots were manually harvested at ripening and the grain obtained was used for YPC determination. The expression of *Psy1-A1* and *Lpx-B1* genes during the grain filling period was monitored in two landraces with contrasting grain yellowness: DW011, *cv*. “Heraldo del Rhin” (b* = 14.18 ± 1.29, Table 3) and DW028, *cv*. “IG-92967” (b* = 21.71 ± 0.55, Table 3).

### 4.2. Detection of QTLs for Carotenoid Content in the RIL Population

The identification of QTLs associated with YI was carried out based on the genetic map previously developed by Giancaspro et al. [35]. QTL detection was performed in QGene 8.3.16 using composite interval mapping (CIM), as proposed by Zeng [48]. The association between marker and trait was considered significant when one or more markers showed a −log10(p) ≥ 3.0, determined by modified Bonferroni correction. The contributions of “02-5B-318” and “Saragolla” were highlighted by a positive and negative sign, respectively.

Graphical representations of linkage groups and QTLs were carried out using MapChart 2.2 software (SolarWinds, Austin, TX, USA). Genes located in the QTL regions were identified by blasting the SNP sequences from Wang et al. [49] against the annotated *Triticum* (NCBI) and contig (URGI) sequences.

In order to pick primer combinations to use for the *Lpx* gene expression analysis, the sequences and the gene models were downloaded from the Svevo [50] and Chinese Spring [51] genome databases. The *Lpx* genes were located on chromosome groups 4 and 5 (4A: TRITD4Av1G195350; 4B: TRITD4Bv1G010710; 5A: TRITD5Av1G200190; 5B: TRITD5Bv1G195300). Primer combinations were designed in the conserved region of the first exon for all genes, considering dissimilarities between the two homoologous copies (Table 9). The genetic materials used for the quantitative real-time PCR analyses were collected from leaves at the seedling stage belonging to the wheat cultivars “Saragolla” and “02-5B-318”, characterized by high and low values of YI, respectively.

Total RNA was extracted from the grain tissue of both genotypes using the RNeasy Plant Mini Kit (QIAGEN^®^) (Germantown, MD, USA) and checked on 1.5% denaturing agarose gel. All RNA samples were the same concentration (1 μg) and were reverse transcribed into double stranded cDNA with a QuantiTect Reverse Trascriptase Kit (QIAGEN^®^). Data were normalized, as previously described by Marcotuli et al. [52], using three reference genes (ADP-RF, RLI, and CDC), which have a stability value, calculated with NormFinder software, of 0.035 [52].

Quantitative real-time PCR was carried out using Cyber^®^ GREEN in the CFX96TM real-time PCR system (BIO-RAD) following the protocol described by Marcotuli et al. [52]. A series of six scalar dilutions were carried out to determine the primer amplification efficiency, while the specificity of the amplicons was confirmed by the following steps: a single band on the agarose gel (2% *w/v*) and a single peak in the melting curves and the sequence of the amplified fragments (3500 Genetic Analyzer, Applied Biosystems).

The qRT-PCR data for genes were derived from the mean values of three independent amplification reactions carried out on the parental lines (“Saragolla” and “02-5B-318”) of the RIL population.

Three reference genes were used to normalize data (cell division control AAA superfamily of ATPases, CDC; ADP-ribosilation factor, ADP-RF; RNase L inhibitor-like protein, RLI [53,54]). All calculations and analyses were performed, as reported by Marcotuli et al. [52], using CFX Manager 2.1 software (Bio-Rad Laboratories), applying the ΔΔCt method. Standard deviations were used to normalize values, and ANOVA and LSD tests were used to underline significant differences between the genotypes.

### 4.3. Psy1-A1 and Lpx-B1 Genotyping in the Mediterranean Panel

Plants were grown in the greenhouse, in triplicate, in a mixture of peat and perlite in a 2:1 ratio, without fertilization, until they reached the two to three leaves stage, in order to extract DNA with the CTAB protocol [55], with minor modifications. One hundred milligrams of leaves were crushed to powder in the presence of liquid nitrogen and 1 mL of CTAB buffer and 0.2% β-mercaptoethanol and 2% PVP were added. The mixture was incubated at 65 °C for 30 min, mixing it gently every 5 min. Then, 1 mL of chloroform and isoamyl alcohol in a ratio of 24:1 were added, mixed by vortexing, and centrifuged at 13,000 rpm for 10 min. Then, the upper phase was extracted and mixed with an equal volume of isopropanol and incubated overnight at −20 °C. Then, the mixture was centrifuged at 13,000 rpm at 4 °C for 10 min and the pellets were air dried at 37 °C, before adding 50 µL of ultrapure water (Invitrogen) (Waltham, MA, USA). Each DNA sequence was quantified by QUBIT 3.0 (Life Technologies) (Carlsbad, CA, USA) and the integrity was checked by agarose electrophoresis.

Regarding *Psy1-A1*, 112 out of 128 genotypes used in this study were previously characterized by Campos et al. [30]. The rest of the population was genotyped with the marker PSY1-A1_STS (Table 5) and PCR conditions used by Singh et al. [29], allowing the discrimination between the alleles *Psy1-A1a, Psy1-A1l*, and *Psy1-A1o*, using the previously extracted DNA.

In order to amplify the *Lpx-B1* genes and different alleles [4,32], the primer pairs in Table 5 were employed in this study. The PCR conditions for every sequence were as follows: *Lpx-B1.1a* (HM126471) and *Lpx-B1.1b* (HM126473) (there is a 74 bp difference between both amplicons that allows differentiation in agarose electrophoresis): one cycle of 1 min at 94 °C; 35 cycles of 30 s at 94 °C followed by 10 s at 68 °C, 1 min 25 s at 72 °C, and one cycle of 7 min at 72 °C. *Lpx-B1.1c* (HM126475): one cycle of 1 min at 94 °C; 35 cycles of 30 s at 94 °C followed by 20 s at 67 °C, 1 min 35 s at 72 °C, and one cycle of 7 min at 72 °C. *Lpx-B1.2* (HM126472) and *Lpx-B1.3* (HM126469) (there is a 76 bp difference between both amplicons that allows differentiation in agarose electrophoresis): one cycle of 1 min at 94 °C, 35 cycles of 30 s at 94 °C followed by 10 s at 62 °C, 1 min 50 s at 72 °C, and one cycle of 7 min at 72 °C. Amplicon identity was checked by sequencing at Macrogen Inc. (Korea). Haplotypes were named following Verlotta et al. [32] nomenclature.

### 4.4. Psy1 and Lpx-1 Gene Expression during Grain Filling

The expression of *Psy1* and *Lpx-1* genes during grain filling was monitored through qPCR. For total RNA extraction, 100 mg of grain tissue were used at each grain developmental stage from 14 DPA to 49 DPA, in triplicate, for genotypes DW011 (low yellowness) and DW028 (high yellowness). RNA extraction was performed based upon the Furtado [56] protocol, with some modifications. The two-step RNA extraction involved the use of TRIzol™ Reagent (ThermoFisher Scientific, Carlsbad, - CA, USA), and the NucleoSpin^®^ RNA Plant Kit (Macherey-Nagel, Düren, Germany). Firstly, seed tissue was pulverized in a mortar and pestle in the presence of liquid nitrogen and 1.5 mL of TRIzol Reagent™ were added. The samples were then centrifuged at 12,000× *g* and 4 °C for 10 min. Subsequently, the upper phase (~750 µL) was recovered, mixed with 300 µL of chloroform and agitated through inversion for 3 min, and then centrifuged for 15 min at 12,000× *g* and 4 °C. The upper phase was then recovered in a new tube, and from this step onwards, the NucleoSpin^®^ RNA Plant Kit was used according to the manufacturer’s instructions. Finally, RNA was eluted with 30 µL RNAse-free H_2_O. RNA integrity was verified on 2% agarose gel and visualized with GelRed^®^ staining (Biotium (company name), Fermont, CA, USA), and RNA was quantified using the NanoDrop 2000 spectrophotometer (ThermoFisher Scientific (company name), Carlsbad, CA, USA). First strand cDNA synthesis was obtained from 4 µg of total RNA using SuperScript™ II (ThermoFisher Scientific (company name), Carlsbad, CA, USA) and oligo dT primers, following the manufacturer’s instructions.

The relative expression analyses were carried out using the SensiFast^TM^ SYBR^®^ No-ROX kit (Bioline (company name), United Kingdom). For each reaction, 50 ng of cDNA were used and tested for the expression of the genes *Psy1* (primers qrtPsy1_F and qrtPsy1_R, Table 5) and *Lpx-1* (primers qrtLpx_B1_F and qrtLpx_B1_R, Table 5) using the ADP ribosylation factor as a reference gene [53]. All primers were designed using the Amplifix and the NCBI primer blast, and they are further characterized in Table 5. The efficiency and CT value of each reaction were determined using LinRegPCR V2018.0 [57], using a common threshold and an individual fit. The relative expression of each gene was calculated using the Pfaffl method [58] with a calibrator value of 0, and then the relative expression was divided by the lower value in each group of data. Three biological and three technical replicates were used at each developmental stage.

### 4.5. Yellowness Determination

Yellow intensity was assessed in the two germplasm panels in wholegrain flour through the yellow index (YI) [11]. L*, a* and b* coordinates in the Munsell color system were taken using D65 lightning. A reflectance colorimeter (CR-400, Konica-Minolta, Japan) equipped with a filter tri-stimulate system was used for these determinations.

### 4.6. Statistical Analyses

Statistical analyses were performed using GraphPad Prism version 8.2.1. GenStat (version 18, VSN International Ltd., Hemel Hempstead, UK), which was used to carry out the ANOVA in the RIL population on the YI to identify how much of the variation was attributed to genotype. Broad-sense heritability (h^2^_B_), was estimated, accounting the genetic variance, σ^2^_G_, the genotype x environment interaction variance, the residual variance, the number of environments (in this case three), and the number of replicates per line (in this case three replicates) [52,59].

## 5. Conclusions

Semolina yellowness is a key aspect determining durum wheat end products. Understanding the synthesis and degradation of carotenoid pigments is a vital aspect of durum wheat breeding programs aiming to produce superior cultivars with higher yellowness. In the current study, eight QTLs related to yellowness were identified, with different degrees of involvement in YI variation, and *Lpx* was linked to a QTL found in chromosome 4B, which showed lower expression in a high yellowness cultivar in comparison to a low yellowness wheat.

The *Lpx-B1* characterization in a landrace and modern wheat Mediterranean population allowed the identification of novel allele combinations (haplotypes IV and V), and by adding a *Psy1-A1* characterization, 11 different combinations of groups appeared, revealing different levels of YI, the modern group being the one with higher levels of yellowness, harboring *Psy1-A1l*, *Lpx-B1.1a*, and *Lpx-B1.2*. Besides, by comparing high and low yellowness landraces, it is possible to suggest that, during grain development, *Psy1* plays a major role contributing to semolina yellowness, and *Lpx* is suggested to be more predominant at post-harvest stages and during pasta making. The practical applications of our work in durum wheat breeding programs relate to the use of marker-assisted selection for *Psy1* alleles showing higher transcript abundance and greater semolina yellowness, whose genotypes could be specifically evaluated and selected at 42 DPA. In addition, studying whether the high yellowness *Psy1* alleles and transcript abundance could be identified at earlier vegetative stages (i.e., leaves) of the plant would be very useful to reduce the selection times without waiting for grain filling. Furthermore, the utilization of biotechnological approaches to elevate the *Psy1* transcripts at late grain development stages would be valuable to potentially increase semolina yellowness. Finally, this study highlights the applicability of markers linked to the *Lpx-1* gene in marker-assisted selection programs.

## Figures and Tables

**Figure 1 ijms-21-04669-f001:**
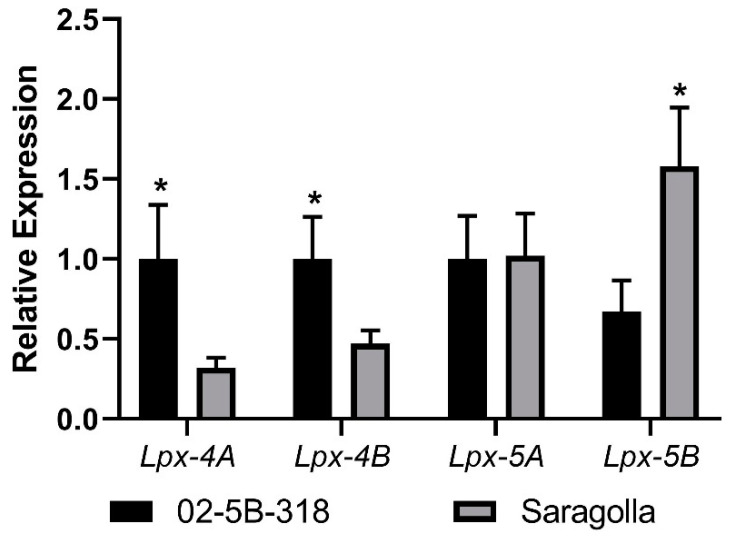
Lipoxygenase (*Lpx*) gene expression on leaves. qPCR was performed on the *Lpx* genes mapped on chromosomes 4A, 4B, 5A, and 5B in leaves of the bread wheat accession “02-5B-318” (low YI) and the durum wheat *cv.* “Saragolla” (high YI). ANOVA and Fisher’s least significant difference (LSD) tests were used to underline significant differences between the genotypes, * *p* < 0.05.

**Figure 2 ijms-21-04669-f002:**
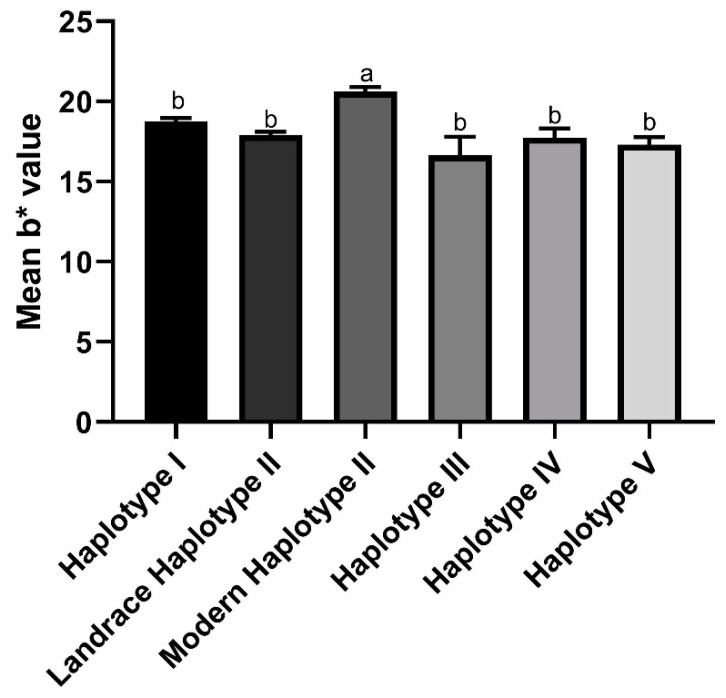
*Lpx-B1* haplotype b* values. Genotypes from the Mediterranean panel were grouped according to their *Lpx-B1* haplotype and a distinction was made between modern cultivars and landraces. One-way ANOVA and Tukey’s post hoc test were performed to find significant differences (*p* < 0.05).

**Figure 3 ijms-21-04669-f003:**
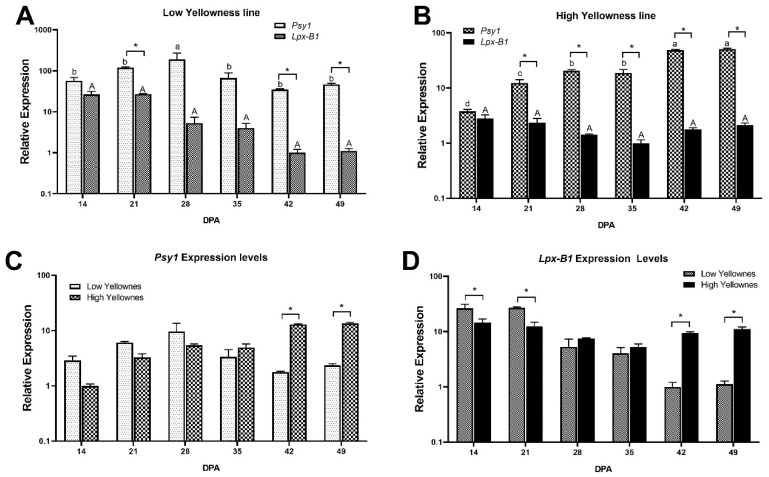
Relative expression of *Psy1* and *Lpx-B1* genes during the grain filling stage in high and low yellowness Mediterranean genotypes. (**A**), relative expression in a low yellowness genotype during grain filling. (**B**), relative expression in a high yellowness genotype during grain filling. (**C**), comparison of the relative expression of *Psy1* between high and low yellowness genotypes during grain filling. (**D**), comparison of the relative expression of *Lpx-B1* in high and low yellowness genotypes. Two-way ANOVA and Tukey’s post hoc test were performed (*p* < 0.05). Upper and lower case are used to compare DPAs for each gene or genotype, an asterisk is used to denote statistical significance between conditions at each DPA. DPA, days post anthesis. All bars represent the mean of three biological and technical replicates (*n* = 9).

**Table 1 ijms-21-04669-t001:** Yellow index (YI ± SE) values of the parental lines and the Recombinant Inbred Line (RIL) population grown in Valenzano for three years. C.V., coefficient of variation; σ^2^_G_, genetic variance; h^2^_B_, broad-sense heritability.

	Year			Mean
	2015	2016	2017	
Parent “02-5B-318”	10.06 ± 0.27	10.37 ± 0.51	10.39 ± 0.25	10.29 ± 0.33
Parent “Saragolla”	13.89 ± 0.11	14.21 ± 0.26	14.80 ± 0.42	14.25 ± 0.37
RIL mean	12.79 ± 1.03	12.75 ± 0.94	13.41 ± 1.71	13.09 ± 1.23
RIL range	10.77–16.15	11.13–17.52	11.78–17.64	
C.V.	3.26	3.01	3.06	
σ^2^_G_	1.02	1.06	0.98	
h^2^_B_	0.85	0.88	0.87	

**Table 2 ijms-21-04669-t002:** Composite interval mapping results estimated for yellow index (YI) from RIL lines derived from the cross “02-5B-318” x “Saragolla” tested at Valenzano in 2015, 2016, and 2017.

Chr	Linkage Group	Marker Interval	Associated Marker	Associated Candidate Gene	Position (cM)	Valenzano 2015			Valenzano 2016			Valenzano 2017			Mean Across Environments		
						Add	LOD	R^2^	Add	LOD	R^2^	Add	LOD	R^2^	Add	LOD	R^2^
2B	2B-4	IWB12724-IWB11333	IWB32245	-	114.50	−96.0	21.4	59.0	−52.0	9.2	30.0	−71.0	11.8	36.0	−66.0	12.8	38.0
4A	4A-2	IWB12722-IWB14501	IWB68425	-	184.10	-	-	-	-	-	-	−3.0	3.1	11.0	-	-	-
4B	4B-2	IWB73832-IWB11928	IWB73831	*Lpx*	2.00	-	-	-	-	-	-	20.0	5.0	10.0	-	-	-
4B	4B-3	IWB71402-IWB53932	IWB15007	*-*	35.50	36.0	4.6	18.0	-	-	-	-	-	-	-	-	-
4B	4B-4	IWA6397-IWB44213	IWB7473	-	38.70	-	-	-	-	-	-	35.0	3.6	13.0	32.0	3.8	13.0
5A	5A-3	IWB65257-IWB35711	IWA850	-	46.10	-	-	-	39.0	5.7	20.0	-	-	-	-	-	-
5A	5A-4	IWA1258-IWB72888	IWA1258	-	0.00	41.0	5.7	21.0	34.0	4.4	15.0	35.0	3.5	12.0	33.0	3.9	14.0
7A	7A-2	IWB73689-IWB25891	IWB72199	-	75.20	36.0	4.5	17.0	33.0	4.3	15.0	45.0	5.7	19.0	42.0	6.0	20.0

**Table 3 ijms-21-04669-t003:** Molecular characterization for *Phytoene synthase 1* (*Psy1-A1*) and *Lipoxygenase 1* (*Lpx-B1*)*,* and yellow index value of the genotypes in the Mediterranean panel. See Table 8 for code interpretation.

Code	*Psy1-A1*	*Lpx-B1.1a*	*Lpx-B1.1b*	*Lpx-B1.1c*	*Lpx-B1.2*	*Lpx-B1.3*	*Lpx-B1* Haplotype	b* Pirque 2017	b* Chillán 2017	b* Chillán 2018	*Psy1-A1/Lpx-B1* Allele Combination
DW024	a		**✓**			**✓**	I	14.45	13.84	13.11	1
DW113	a		**✓**			**✓**	I	15.15	16.04	15.24	1
DW001	l		**✓**			**✓**	I	19.30	21.04	21.65	2
DW002	l		**✓**			**✓**	I	16.89	18.71	20.59	2
DW006	l		**✓**			**✓**	I	18.78	18.79	20.09	2
DW014	l		**✓**			**✓**	I	18.80	18.64	20.10	2
DW019	l		**✓**			**✓**	I	18.20	17.03	15.23	2
DW020	l		**✓**			**✓**	I	21.74	21.41	20.93	2
DW022	l		**✓**			**✓**	I	18.80	18.81	18.52	2
DW025	l		**✓**			**✓**	I	13.88	13.38	12.38	2
DW032	l		**✓**			**✓**	I	20.34	21.70	22.70	2
DW034	l		**✓**			**✓**	I	19.87	19.54	20.62	2
DW037	l		**✓**			**✓**	I	17.27	17.42	18.99	2
DW043	l		**✓**			**✓**	I	19.02	18.34	23.12	2
DW045	l		**✓**			**✓**	I	17.44	17.52	16.98	2
DW056	l		**✓**			**✓**	I	19.39	18.25	21.84	2
DW058	l		**✓**			**✓**	I	16.57	16.42	17.39	2
DW061	l		**✓**			**✓**	I	18.95	19.45	20.99	2
DW067	l		**✓**			**✓**	I	16.96	17.66	17.61	2
DW071	l		**✓**			**✓**	I	17.81	18.36	18.19	2
DW074	l		**✓**			**✓**	I	21.27	20.15	21.10	2
DW083	l		**✓**			**✓**	I	18.93	17.34	14.87	2
DW084	l		**✓**			**✓**	I	14.60	18.60	17.23	2
DW090	l		**✓**			**✓**	I	14.57	14.76	15.96	2
DW092	l		**✓**			**✓**	I	18.34	18.97	18.89	2
DW093	l		**✓**			**✓**	I	16.65	16.44	18.49	2
DW096	l		**✓**			**✓**	I	17.31	20.29	18.39	2
DW099	l		**✓**			**✓**	I	20.77	22.06	23.82	2
DW103	l		**✓**			**✓**	I		18.41	19.29	2
DW115	l		**✓**			**✓**	I	16.33	15.85	16.32	2
DW121	l		**✓**			**✓**	I	15.89	16.98	16.94	2
DW124	l		**✓**			**✓**	I	24.17	23.17	24.38	2
DW126	l		**✓**			**✓**	I	16.77	13.23	16.91	2
DW127	l		**✓**			**✓**	I	16.59	18.49	18.16	2
DW130	l		**✓**			**✓**	I	18.91	17.91	17.04	2
DW139	l		**✓**			**✓**	I	22.78	22.87	22.78	2
DW149	l		**✓**			**✓**	I	22.25	21.93	22.74	2
DW160	l		**✓**			**✓**	I	25.70	25.44	25.78	2
DW162	l		**✓**			**✓**	I	17.17	14.98	18.64	2
DW166	l		**✓**			**✓**	I	16.59	17.27	17.94	2
DW172	l		**✓**			**✓**	I	19.36	17.13	21.12	2
DW012	o		**✓**			**✓**	I	17.41	18.36	18.94	3
DW015	o		**✓**			**✓**	I	20.23	20.00	20.89	3
DW017	o		**✓**			**✓**	I	23.71	23.63	24.06	3
DW028	o		**✓**			**✓**	I		21.32	22.79	3
DW038	o		**✓**			**✓**	I	16.36	14.56	16.20	3
DW080	o		**✓**			**✓**	I	16.95	18.37	20.49	3
DW082	o		**✓**			**✓**	I	18.38	19.82	18.38	3
DW086	o		**✓**			**✓**	I	14.61	12.59	14.04	3
DW129	o		**✓**			**✓**	I	23.41	21.60	21.73	3
DW009	a	**✓**			**✓**		II	18.91	18.29	17.96	4
DW011	a	**✓**			**✓**		II	12.94	16.76	12.83	4
DW013	a	**✓**			**✓**		II		18.54	15.89	4
DW060	a	**✓**			**✓**		II	19.69	19.04	21.45	4
DW069	a	**✓**			**✓**		II	14.94	15.99	15.95	4
DW070	a	**✓**			**✓**		II	14.44	16.46	18.51	4
DW072	a	**✓**			**✓**		II	13.54	12.40	10.32	4
DW110	a	**✓**			**✓**		II	14.18	12.77	14.66	4
DW134	a	**✓**			**✓**		II		14.56	15.18	4
DW136	a	**✓**			**✓**		II		16.29	14.03	4
DW138	a	**✓**			**✓**		II	17.68	18.83	18.57	4
DW148	a	**✓**			**✓**		II	14.60	15.19	17.52	4
DW154	a	**✓**			**✓**		II		16.16	15.91	4
DW191	a	**✓**			**✓**		II	17.84	18.47	19.00	4
DW003	l	**✓**			**✓**		II	14.71	12.98	14.59	5
DW004	l	**✓**			**✓**		II	16.61	18.21	15.96	5
DW005	l	**✓**			**✓**		II	16.45	16.71	17.20	5
DW010	l	**✓**			**✓**		II		18.90	15.92	5
DW016	l	**✓**			**✓**		II	17.59	19.63	18.07	5
DW018	l	**✓**			**✓**		II	22.26	20.09	20.29	5
DW021	l	**✓**			**✓**		II	19.02	20.94	18.80	5
DW027	l	**✓**			**✓**		II	17.23	18.35	19.73	5
DW033	l	**✓**			**✓**		II	17.76	18.22	18.48	5
DW041	l	**✓**			**✓**		II	18.66	18.63	20.02	5
DW042	l	**✓**			**✓**		II	16.26	15.42	15.14	5
DW044	l	**✓**			**✓**		II	16.91	17.62	19.78	5
DW046	l	**✓**			**✓**		II	16.03	17.01	17.19	5
DW048	l	**✓**			**✓**		II	16.65	16.84	17.79	5
DW052	l	**✓**			**✓**		II	18.32	16.38	19.72	5
DW054	l	**✓**			**✓**		II	18.41	20.04	21.38	5
DW062	l	**✓**			**✓**		II	19.19	19.57	20.15	5
DW068	l	**✓**			**✓**		II	16.52	18.67	19.59	5
DW091	l	**✓**			**✓**		II	20.19	21.24	20.03	5
DW094	l	**✓**			**✓**		II	14.28	14.29	16.99	5
DW095	l	**✓**			**✓**		II	16.25	19.70	19.68	5
DW104	l	**✓**			**✓**		II	20.19	20.92	21.87	5
DW117	l	**✓**			**✓**		II	20.47	21.83	18.68	5
DW128	l	**✓**			**✓**		II		21.38	20.81	5
DW131	l	**✓**			**✓**		II	15.32	15.80	16.96	5
DW132	l	**✓**			**✓**		II	20.85	22.23	21.63	5
DW133	l	**✓**			**✓**		II		22.08	21.94	5
DW137	l	**✓**			**✓**		II		21.08	21.09	5
DW144	l	**✓**			**✓**		II	21.07	20.71	20.96	5
DW145	l	**✓**			**✓**		II	20.04	20.73	21.04	5
DW146	l	**✓**			**✓**		II		19.56	20.17	5
DW150	l	**✓**			**✓**		II	21.72	22.59	21.82	5
DW151	l	**✓**			**✓**		II	19.97	20.75	19.68	5
DW158	l	**✓**			**✓**		II	15.26	16.90	15.98	5
DW161	l	**✓**			**✓**		II	17.05	18.35	18.18	5
DW163	l	**✓**			**✓**		II	17.76	15.35	19.48	5
DW165	l	**✓**			**✓**		II	15.32	15.87	18.20	5
DW167	l	**✓**			**✓**		II	16.93	16.32	15.03	5
DW168	l	**✓**			**✓**		II	14.30	14.75	17.65	5
DW170	l	**✓**			**✓**		II	15.68	15.25	16.79	5
DW171	l	**✓**			**✓**		II	17.57	18.54	19.06	5
DW174	l	**✓**			**✓**		II	18.45	18.86	21.54	5
DW175	l	**✓**			**✓**		II	20.01	20.17	21.70	5
DW176	l	**✓**			**✓**		II	19.48	20.32	20.87	5
DW177	l	**✓**			**✓**		II	20.58	20.70	21.53	5
DW187	l	**✓**			**✓**		II	21.73	21.68	21.33	5
DW189	l	**✓**			**✓**		II	18.83	21.21	20.10	5
DW190	l	**✓**			**✓**		II	23.49	22.93	23.33	5
DW192	l	**✓**			**✓**		II	19.72	22.65	20.16	5
DW023	o	**✓**			**✓**		II	20.75	21.73	21.78	6
DW078	o	**✓**			**✓**		II	22.54	23.17	21.33	6
DW085	o	**✓**			**✓**		II	15.06	13.65	14.31	6
DW098	o	**✓**			**✓**		II	16.97	17.91	18.14	6
DW116	o	**✓**			**✓**		II	15.75	16.82	16.90	6
DW152	o	**✓**			**✓**		II	16.29	18.05	18.34	6
DW059	l			**✓**	**✓**		III	14.05	14.98	13.27	7
DW008	o			**✓**	**✓**		III	18.45	19.84	19.24	8
DW047	l		**✓**		**✓**		IV	16.34	15.98	16.54	9
DW122	l		**✓**		**✓**		IV	15.52	18.46	18.00	9
DW147	l		**✓**		**✓**		IV	19.10	19.56	20.24	9
DW035	l	**✓**				**✓**	V	15.47	15.45	17.32	10
DW040	l	**✓**				**✓**	V	20.00	18.36	20.38	10
DW111	l	**✓**				**✓**	V	17.06	17.87	17.42	10
DW114	l	**✓**				**✓**	V	15.31	16.09	16.86	10

**Table 4 ijms-21-04669-t004:** Allele variants of the *Psy1-A1* gene identified in the Mediterranean durum wheat panel and associated b* values.

Allele	Number of Genotypes	Frequency (%)	n^a^	Minimum	Maximum	Mean ± SE^b^
*Psy1-A1a*	16	12.4	44	10.32	21.45	16.00 ± 0.35^B^
*Psy1-A1l*	97	75.2	282	12.38	25.78	18.66 ± 0.15^A^
*Psy1-A1o*	16	12.4	48	12.59	24.06	18.89 ± 0.43^A^

^a^ Data from Table 3, b* values for Pirque 2017, Chillán 2017, and Chillán 2018. ^b^ Different uppercase letters correspond to significantly different values after one-way ANOVA and Tukey’s test (*p* < 0.05).

**Table 5 ijms-21-04669-t005:** List of primers used to genotype the Mediterranean panel for *Psy1-A1* and *Lpx-B1*.

Primer Name	Sequence (5′ → 3′)	Gene/Allele	Product Length (bp)	Reference
Psy1-A1_STS_R	GTG GAT ATT CCC TGT CAG CATC	*Psy1-A1o - Psy1-A1l - Psy1-A1a*	897 - 1089 - 1776	Singh et al. [29]
Psy1-A1_STS_F	GCC TCC TCG AAG AAC ATC CTC			
qrtPsy1_F	GCGAGGGGTGACTGAGCTT	*Psy1*	117	Rodriguez-Suarez et al. [36]
qrtPsy1_R	CTCTTGGTGAAGTTGTTGTAGTCA			
qrtLpx_B1_F	TCAACTTCGGGCAGTACCCATA	*Lpx-B1*	177	This study
qrtLpx_B1_R	CCAACAGCGAGATGCCAATGAT			
Lpx-B1.1a/1b Forward	GCA GGC GCT GGA AAG CAA CAG GC	*Lpx-B1.1a - Lpx-B1.1b*	1320 - 1246	Verlotta et al. [32]
Lpx-B1.1a/1b Reverse	GCG CTC TAA CTC CGC GTA CTC G			
Lpx-B1.1c_Forward	CCA AGA TGA TAC TGG GCG GGC	*Lpx-B1.1c*	1558	
Lpx-B1.1c_Reverse	CGC CGC CTT GCC GTG GTT GG			
Lpx-B1.2/1.3 Forward	GAA CCG AGA GGT GAG AGC GTG CCT GAT C	*Lpx-B1.2 - Lpx-B1.3*	1785 - 1709	This study
Lpx-B1.2/1.3 Reverse	GTG GTC GGA GGT GTT GGG GTA GAG C			

**Table 6 ijms-21-04669-t006:** *Lpx-B1* haplotypes identified in the Mediterranean panel and average b* values across Pirque 2017, Chillán 2017, and Chillán 2018. Differences in b* values were not statistically significant (*p* > 0.05).

Haplotype	Allelic Combination	N. of Genotypes	Frequency (%)	n^a^	Minimum	Maximum	Mean ± SE
I	*Lpx-B1.1b* and *Lpx-B1.3*	50	39.1	149	12.38	25.78	18.74 ± 0.23
II	*Lpx-B1.1a* and *Lpx-B1.2*	69	53.9	198	10.32	23.49	18.28 ± 0.18
III	*Lpx-B1.1c* and *Lpx-B1.2*	2	1.6	6	13.27	19.84	16.64 ± 1.17
IV	*Lpx-B1.1b* and *Lpx-B1.2*	3	2.3	9	15.52	20.24	17.75 ± 0.57
V	*Lpx-B1.1a* and *Lpx-B1.3*	4	3.1	12	15.31	20.38	17.30 ± 0.48

^a^ Data from Table 3, b* values for Pirque 2017, Chillán 2017, and Chillán 2018.

**Table 7 ijms-21-04669-t007:** Frequency of the *Psy1-A1* + *Lpx-B1* allele combinations identified in the Mediterranean panel and yellow index (b* values) across Pirque 2017, Chillán 2017, and Chillán 2018.

Allelic Combination	Composition	Number of Genotypes	Frequency (%)	n^a^	Minimum	Maximum	Mean ± SE^b^
1	Haplotype I + *Psy1-A1a*	2	1.6	6	13.11	16.04	14.64 ± 0.43^E^
2	Haplotype I + *Psy1-A1l*	39	30.5	116	12.38	25.78	18.83 ± 0.25^C^
3	Haplotype I + *Psy1-A1o*	9	7.0	27	12.59	24.06	19.25 ± 0.61^ABC^
4	Haplotype II + *Psy1-A1a*	14	10.9	38	10.32	21.45	16.22 ± 0.39^E^
5 Landraces	Haplotype II + *Psy1-A1l*	41	32.0	118	12.98	22.59	18.41 ± 0.2^BCD^
5 Modern	Haplotype II + *Psy1-A1l*	8	6.3	24	18.45	23.49	20.89 ± 0.28^A^
6	Haplotype II + *Psy1-A1o*	6	4.7	18	13.65	23.17	18.31 ± 0.69^BCDE^
7	Haplotype III + *Psy1-A1l*	1	0.8	3	13.27	14.98	14.10 ± 0.49^DE^
8	Haplotype III + *Psy1-A1o*	1	0.8	3	18.45	19.84	19.18 ± 0.40^ABCDE^
9	Haplotype IV + *Psy1-A1l*	3	2.3	9	15.52	20.24	17.75 ± 0.57^BCDE^
10	Haplotype V + *Psy1-A1l*	4	3.1	12	15.31	20.38	17.30 ± 0.48^BCDE^

^a^ Data from Table 3, b* values for Pirque 2017, Chillán 2017, and Chillán 2018. ^b^ Means with different uppercase letters correspond to significantly different values after one-way ANOVA and Tukey’s post hoc test (*p* < 0.05).

**Table 8 ijms-21-04669-t008:** Genotypes included in the Mediterranean panel. LR, landraces.

Code	Type	Genotype	Origin	Code	Type	Genotype	Origin
DW001	LR	Arisnegro de Tenerife	Spain	DW091	LR	9923	Lebanon
DW002	LR	Basto Duro	Spain	DW092	LR	9929	Lebanon
DW003	LR	Blanco de Corella	Spain	DW093	LR	9935	Lebanon
DW004	LR	Blanquillo	Spain	DW094	LR	Abu Fashit	Israel
DW005	LR	Candeal de Salamanca	Spain	DW095	LR	D-2	Egypt
DW006	LR	Colorado de Jerez	Spain	DW096	LR	Maghoussa	Morocco
DW008	LR	Fartó	Spain	DW098	LR	Red Beard	Morocco
DW009	LR	Griego de Baleares	Spain	DW099	LR	Safra Jerash	Jordan
DW010	LR	Gros de Cerdaña	Spain	DW103	LR	Louri AP5	Tunisia
DW011	LR	Heraldo del Rhin	Spain	DW104	LR	Souri	Tunisia
DW012	LR	Pinet	Spain	DW110	LR	1P1	Egypt
DW013	LR	Pisana cañihueca	Spain	DW111	LR	Beladi Rouge	France
DW014	LR	Raspinegro Canario	Spain	DW113	LR	lumillo	France
DW015	LR	Raspinegro de Alcalá Guadaira	Spain	DW114	LR	Tounse	France
DW016	LR	Recio de Almería	Spain	DW115	LR	Trigo Glutinoso	France
DW017	LR	Verdial	Spain	DW116	LR	9918	Lebanon
DW018	LR	Tchirpan	Bulgaria	DW117	LR	Hourah	Lebanon
DW019	LR	Lozen 76	Bulgaria	DW121	LR	Maghoussa Amizmiz	Morocco
DW020	LR	Vroulos	Cyprus	DW122	LR	Muri	Cyprus
DW021	LR	IG-82549	Cyprus	DW124	LR	Harani Auttma	Jordan
DW022	LR	Carlantino	Italy	DW126	LR	Horani Howawi	Jordan
DW023	LR	Cicirelo	Italy	DW127	LR	Zugbieh Sutra	Jordan
DW024	LR	IG-83905	Italy	DW128	LR	Belgrade 9	Serbia
DW025	LR	IG-83920	Italy	DW129	LR	Zoghbiyeh Safra	Jordan
DW027	LR	IG-92895	Algeria	DW130	LR	Etith	Israel
DW028	LR	IG-92967	Algeria	DW131	LR	Juljulith	Israel
DW032	LR	IG-95812	Syria	DW132	LR	248-VII/7	Macedonia
DW033	LR	IG-95841	Syria	DW133	LR	259-VII/12	Macedonia
DW034	LR	IG-95847	Syria	DW134	LR	356-I/9	Montenegro
DW035	LR	IG-95931	Syria	DW136	LR	441-IX/97	Croatia
DW037	LR	IG-96851	Creta	DW137	LR	VII/13-X11	Macedonia
DW038	LR	Alonso	Spain	DW138	LR	VII/18-X24	Macedonia
DW040	LR	Azulejo de Villa del Río	Spain	DW139	LR	Safra Maan	Jordan
DW041	LR	Blancal	Spain	DW144	LR	196/71	Macedonia
DW042	LR	Blanquillón de Boñar	Spain	DW145	LR	1575	Serbia
DW043	LR	Claro de Balazote	Spain	DW146	LR	II/4	Macedonia
DW044	LR	Entrelargo de Montijo	Spain	DW147	LR	Cobros	Morocco
DW045	LR	Farto cañifino	Spain	DW148	LR	Haj Mouline	Morocco
DW046	LR	Rubio de Miajadas	Spain	DW149	LR	26	Jordan
DW047	LR	Rubio de Montijo	Spain	DW150	LR	Mavraani	Greece
DW048	LR	Ruso	Spain	DW151	LR	Rapsani	Greece
DW052	LR	Senatore Capelli	Italy	DW152	LR	Giza 2	Egypt
DW054	LR	Hymera	Italy	DW154	LR	33	Montenegro
DW056	LR	Aziziah 17/45	Italy	DW158	LR	Rubio enlargado d’ Atlemtejo	France
DW058	LR	Balilla Falso	Italy	DW160	LR	Hati	Israel
DW059	LR	Marques	Portugal	DW161	LR	Tripshiro	Libya
DW060	LR	Mindium	Turkey	DW162	LR	2751	Egypt
DW061	LR	Raposinho	Portugal	DW163	LR	JM-3987	Israel
DW062	LR	Reading	Egypt	DW165	LR	MG26429	Egypt
DW067	LR	Anafil	Portugal	DW166	LR	18/71	Serbia
DW068	LR	Espanhol	Portugal	DW167	LR	28	Egypt
DW069	LR	Dezassete	Portugal	DW168	LR	31	Egypt
DW070	LR	Durazio Rijo Glabro	Portugal	DW170	LR	Mishriki	Egypt
DW071	LR	Amarelo Barba Preta	Portugal	DW171	LR	Girgeh	Egypt
DW072	LR	Alentejo	Portugal	DW172	LR	Hamira	Tunisia
DW074	LR	BGE-018192	Turkey	DW174	Modern	Ancalei	Spain
DW078	LR	BGE-018354	Turkey	DW175	Modern	Arment	France
DW080	LR	BGE-019263	Turkey	DW176	Modern	Astigi	Spain
DW082	LR	BGE-019265	Turkey	DW177	Modern	Boabdil	Spain
DW083	LR	BGE019266	Turkey	DW187	Modern	Senadur	Spain
DW084	LR	BGE-019270	Turkey	DW189	Modern	Sula	Spain
DW085	LR	Tremes rijo	Portugal	DW190	Modern	Svevo	Italy
DW086	LR	Lobeiro de grao escuro	Portugal	DW191	Modern	Vitron	Spain
DW090	LR	Zoco Yebel Hebil	Morocco	DW192	Modern	Vitronero	Spain

**Table 9 ijms-21-04669-t009:** Primer pairs used for *Lpx* gene expression analysis in the RIL population.

Gene Name	Primer Orientation	Sequence (5′ → 3′)	Product Length (bp)
*Lpx-4A*	Fw	CAG TTC CAG ACC ATC CTC GG	229
	Rv	TGT AGG GCA TCT TCA CCG G	
*Lpx-4B*	Fw	ATC CTG TCC AAG CAC TCC TC	240
	Rv	CAG CCC TTT CTC GCC GTC	
*Lpx-5A*	Fw	GAG GTC TGG CAC GCG ATC	171
	Rv	CAC GGT GTG CAT CTT GGG C	
*Lpx-5B*	Fw	CAA GAT GCA GAC GGT GGC	217
	Rv	GGT GAT GGT GAG GAT AAA GGC	

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
