# Peer review of "Phytoene synthase 1 (Psy-1) and lipoxygenase 1 (Lpx-1) Genes Influence on Semolina Yellowness in Wheat Mediterranean Germplasm"

_ijms, 2020, doi:10.3390/ijms21134669_

Round 1

Reviewer 1 Report

Although the authors claim, “This innovative study was conducted…” in line # 94; there is not much novelty in the study.

Routine work with some useful information is provided.

The authors should have tried transcriptome in the selected lines to confirm there are no additional genes involved in the trait.

Author Response

Reviewer 1 wrote:

“Although the authors claim, “This innovative was conducted” in line 94; there is not much novelty in the study”

Author´s response: thank you for the revision of the paper. We do not agree on the point about the lack of innovation of this paper. Abundant studies have been conducted trying to investigate how grain yellow piment content (YPC) of durum wheat affects semolina color and pasta quality, but few analyzing important candidate genes influencing YPC from a QTL mapping, haplotype analysis and gene expression approaches. Additionally, all the papers studying the yellowness of wheat kernel have mainly reported the key role of the Phytoene synthase 1 gene, without considering other putative genes involved in this complex pathway. The data reported here highlighted the importance of other genes in the control of this trait.  In this sense, we are not aware of any other publication that studies Phytoene synthase 1 (Psy-1) and lipoxygenase 1 (Lpx-1) jointly, which are relevant genes used in marker-assisted selection for improving YPC in durum wheat breeding programs. In addition, two new Lpx-1 haplotypes were identified in the IRTA Mediterranean population in our study, named haplotypes IV and V following the nomenclature of Verlotta et al. [2010], and they were reported for the first time in the literature. Consequently, we believe this work is original and it contains several scientific novelties and practical applications that will have international impact. However, we deleted the term “innovative” based on the request of reviewer 1 (please see line 104, revised version).

“Routine work with some useful information is provided”

Author´s response: thank you for this comment.

“The authors should have tried transcriptome in the selected lines to confirm there are no additional involved in the trait”

Author´s response: we completely agree with reviewer 1 that the results found should be complemented using transcriptomic analyses. Actually, the analysis of the transcriptome represents a new step in our work, and in our opinion all the data and results can be fitted in another manuscript. Here we reported different approaches and material to reach the aim.

Reviewer 2 Report

Author identified new allele variants and/or allele combinations significantly affecting YPC in semolina through their effect on synthesis and degradation of carotenoid pigments in grain and semolina. For this purpose, three complementary approaches were undertaken: the identification of QTLs associated to carotenoid content on a RIL (recombinant inbreed line) population, the characterization of a Mediterranean panel of accessions for Psy1 and Lpx-1 genes, and monitoring the expression of Psy1 and Lpx-1 genes during grain filling on two genotypes with contrasting YPC. By comparing high and low yellowness landraces, it is possible to suggest that during grain development, Psy1 plays a major role contributing to semolina yellowness and Lpx is suggested to be more predominant at post-harvest stages and during pasta making processing.

  • The topic is within the scope of this journal. However, the manuscript preparation does not reach to the standards of scientific publication.
  • Abstract need to be improve.
  • Background of the study should be made to very clear. Provide more details of introduction and review of the work.
  • Please speculate about the reasons to the obtained results.
  • In Conclusion, authors should add significance of this research to potential practical application.
  • The whole MS need to be improved.
  • Lack of results, If possible, author needs to add some additional experiments.
  • English writing needs to be improved.

Author Response

Reviewer 2 wrote:

“The topic is within the scope of this journal. However, the manuscript preparation does not reach the standards of scientific publication”

Author´s response: we regret to receive this comment by reviewer 2. If the manuscript is revised in depth, one can realize that the authors worked in two different countries, in Italy and Chile, with trials starting in 2015 and 2017, respectively, which means that this manuscript summarizes our research work of the last 5 years. QTL mapping, haplotype analysis of a diverse Mediterranean durum wheat population and gene expression experiments are a robust approach to investigate YPC, and they were conducted under high scientific standards based upon the methodology used in this study and the publication records of the authors.  

“Abstract need to be improve”

Author´s response: the abstract has been rewritten and improved following the suggestions of the reviewer.

“Background of the study should be made to very clear. Provide more details of introduction and review of the work”

Author´s response: thank you for this comment. We agree with reviewer 2 that this section can be improved by adding more information about the role of research on durum wheat grain quality. In this context, we incorporated the following paragraphs:

  1. Lines 40-46 (revised version): “In the twentieth century, major breeding programs were focused on improving the durum productivity traits of wheat, such as grain yield, biotic and abiotic stress resistance. In this century, the attention on food quality over quantity has switched the research aims at increasing wheat nutritional value estimated through different parameters, like protein content, water absorption and flour color. The latter one is due to the carotenoid pigments, which have an enormous importance for the nutritional value in human health (Sommer and Davidson, 2002)”.
  2. Lines 50-53 (revised version): “particularly considering that the competition in the pasta market has made this trait even more important (Dexter and Marchylo, 2001), especially after the legal ban of the use of artificial coloring in pasta production in certain countries in Europe (Hare, 2006), which has strengthened the role of durum wheat breeding programs for YPC enhancements”.
  3. Lines 55-56 (revised version): “Lutein is a compound that contributes to the organoleptic quality of pasta (yellowness) (Hentschel et al., 2002)”.

“Please speculate about the reasons to the obtained results”

Author´s response: The following sections were added into the Discussion section:

  1. Line 136 (revised version): “Carrera et al. [4] showed how the role of lipoxygenase on carotenoid degradation occurred in the process of pasta making and not during wheat grain development, which is in agreement with our own results”.
  2. Lines 164-167 (revised version): “Further, we can speculate that differential Lpx- 1 gene expression identified during vegetative stage in this study, but not during grain filling, may probably not affect YPC levels, but post-transcriptional mechanisms, durum wheat processing conditions and/or other allelic variants for Lpx-1 do influence semolina yellowness”.
  3. Lines 202-203 (revised version): “Molecular markers associated to Lpx-1 are currently being effectively used for marker-assisted selection in durum wheat breeding programs to improve YPC in different parts of the world, including the United States (Borrelli and Trono, 2016), and Canada (Randhawa et al., 2013).”

“In Conclusion, authors should add significance of this research to potential practical application”

Author´s response: The following paragraph was incorporated into the Conclusions part:

“The practical applications of our work in durum wheat breeding programs relate to the use of marker-assisted selection for Psy-1 alleles showing higher transcript abundance and greater semolina yellowness, whose genotypes could be specifically evaluated and selected at 42 DPA. In addition, studying whether the high-yellowness Psy-1 alleles and transcript abundance could be identified at earlier vegetative stages (i.e., leaves) of the plant would be very useful to reduce the selection times without waiting for grain filling. Further, the utilization of biotechnological approaches to elevate the Psy-1 transcripts at late grain development stage would be valuable to potentially increase semolina yellowness. Finally, this study highlights the applicability of markers linked to the Lpx-1 gene in marker-assisted selection programs.”

“The whole MS need to be improved”

Author´s response: Improvements have been made in the Abstract, Introduction, Discussion and Conclusion sections, as previously indicated.

“Lack of results, If possible, author needs to add some additional experiments”

Author´s response: at the moment, considering the worldwide situation due to the pandemia, we are not able to provide additional experiments to add to this manuscript. We have significantly improved the quality of the present manuscript, following the suggestions of the reviewers.

“English writing needs to be improved”

Author´s response: We have reviewed again the English writing carefully throughout the manuscript and have made minor changes.

References

Borrelli, G.M.; Trono, D. Molecular Approaches to Genetically Improve the Accumulation of Health-Promoting Secondary Metabolites in Staple Crops-A Case Study: The Lipoxygenase-B1 Genes and Regulation of the Carotenoid Content in Pasta Products. Int J Mol Sci 2016, 17, doi:10.3390/ijms17071177.

Dexter, J. E., and B.A. Marchylo. 2000. Recent Trends in Durum Wheat Milling and Pasta Processing: Impact on Durum Wheat Quality Requirements. In: Proceedings of the international workshop on durum wheat, semolina, and pasta quality: recent achievements and new trends, Institute National de la Recherche Agronomique. Montpellier, France, 27 November 2000.

Hentschel, V., K. Kranl, J. Hollmann, M.G. Lindhauer, V. Bohm, and R. Bitsch. 2002. Spectrophotometric determination of yellow pigment content and evaluation of carotenoids by high-performance liquid chromatography in durum wheat grain. J. Agr. Food Chem. 50:6663-6668.

Randhawa, H.S.; Asif, M.; Pozniak, C.; Clarke, J.M.; Graf, R.J.; Fox, S.L.; Humphreys, D.G.; Knox, R.E.; DePauw, R.M.; Singh, A.K., et al. Application of molecular markers to wheat breeding in Canada. Plant Breeding 2013, 10.1111/pbr.12057, n/a-n/a, doi:10.1111/pbr.12057.

Sommer, A., and Davidson, F. R.. 2002. Assessment and control of vitamin A deficiency: The annecy accords. J. Nutr. 132, 2845S-2850S. doi: 10.1093/jn/132.9.2845S.

Round 2

Reviewer 2 Report

Requested corrections were carried out.